# Stakeholder perspectives on contributors to delayed and inaccurate diagnosis of cardiovascular disease and their implications for digital health technologies: a UK-based qualitative study

Kamilla Abdullayev ![ORCID],[1] Olivia Gorvett,[1] Anna Sochiera,[1] Lynn Laidlaw,[2] Timothy Chico,[3] Matthew Manktelow,[4] Oliver Buckley,[5] Joan Condell,[4] Richard Van Arkel,[6] Vanessa Diaz,[7,8] Faith Matcham[1]

For numbered affiliations see end of article.

**Correspondence to**
Kamilla Abdullayev;
kga21@sussex.ac.uk

## ABSTRACT

**Objective** The aim of this study is to understand stakeholder experiences of diagnosis of cardiovascular disease (CVD) to support the development of technological solutions that meet current needs. Specifically, we aimed to identify challenges in the process of diagnosing CVD, to identify discrepancies between patient and clinician experiences of CVD diagnosis, and to identify the requirements of future health technology solutions intended to improve CVD diagnosis.

**Design** Semistructured focus groups and one-to-one interviews to generate qualitative data that were subjected to thematic analysis.

**Participants** UK-based individuals (N=32) with lived experience of diagnosis of CVD (n=23) and clinicians with experience in diagnosing CVD (n=9).

**Results** We identified four key themes related to delayed or inaccurate diagnosis of CVD: symptom interpretation, patient characteristics, patient–clinician interactions and systemic challenges. Subthemes from each are discussed in depth. Challenges related to time and communication were greatest for both stakeholder groups; however, there were differences in other areas, for example, patient experiences highlighted difficulties with the psychological aspects of diagnosis and interpreting ambiguous symptoms, while clinicians emphasised the role of individual patient differences and the lack of rapport in contributing to delays or inaccurate diagnosis.

**Conclusions** Our findings highlight key considerations when developing digital technologies that seek to improve the efficiency and accuracy of diagnosis of CVD.

## STRENGTHS AND LIMITATIONS OF THIS STUDY

⇒ Inclusion of both patients and clinicians in our study is a strength as it allows a more complete understanding of the barriers preventing accurate and efficient cardiovascular disease (CVD) diagnosis and comparison between these groups.

⇒ Decentralised recruitment meant the study included a range of individuals from across the UK with a variety of different CVD and professional experiences.

⇒ Conversely, use of online recruitment platforms and snowball convenience sampling to recruit participants may have produced a biased sample who are more involved in their own healthcare, more familiar with technology and more inclined towards new technological developments.

⇒ The lack of ethnic diversity within participants means our findings may not be representative of all groups.

## INTRODUCTION

A total of 6.8 million people have cardiovascular disease (CVD) in England, and 25% of deaths in the UK are caused by CVD.[1] The economic and social burden of CVD continues to increase globally.[2] CVDs cost the UK economy around £7.4 billion annually in healthcare costs, rising to £15.8 billion when considering wider economic costs.[3] CVD high excess primary and secondary care costs are doubled when patients have previously suffered cardiovascular events.[4] CVDs also incur severe psychological and social consequences which extend beyond the patient to their families and support network.[5,6]

Although the negative impact of CVDs is well-known, recent evidence suggests only half of patients had a primary care consultation prior to their diagnosis and only 24% of patients experienced the recommended pathway to diagnosis.[7] Moreover, a recent systematic review revealed misdiagnosis of heart failure ranges from 16% in patients discharged from hospital to 68% in primary care.[8] Delayed and inaccurate diagnoses are

common in specific CVDs, such as amyloid cardiomyopathy[9] and pulmonary embolism.[10] Inequitable access to healthcare, symptom recognition disparities, structural influences on provision of timely and high-quality care, and bias among clinicians may all contribute to delays in CVD diagnosis.[11] There are also likely to be issues related to healthcare access and diagnosis following the COVID-19 pandemic, which led to reduced CVD admissions and increased CVD mortality.[12] Delayed or misdiagnosis contributes to inappropriate treatments or unnecessary evaluations.[13 14] Adverse outcomes such as hospitalisation or death may result from effective treatment not being received until the disease is more advanced.[15] For example, a missed diagnosis of heart failure is associated with increased hospital readmission rates[16] and with a twofold increased risk of death.[17] Avoiding misdiagnosis and reducing diagnosis delays are critical for improving patient outcomes and reducing healthcare costs. Technological advances and increased pressures on healthcare systems are driving great interest in the use of digital health technologies in healthcare. The pandemic prompted a sharp rise in the use of remote measurement technologies in healthcare,[18–20] while around 40% of the UK population uses a wearable that monitors health measures such as sleep, activity and heart rate.

A range of potential digital technologies have been proposed to address the challenges in diagnosing CVD.[21] One such technology is a 'digital twin'.[22–24] Digital twins address the problem that healthcare professionals today are required to assimilate huge amounts of data on each patient (including description of symptoms, medical and medication history, laboratory tests, X-rays and other imaging, physiological measures such as heart rate, blood pressure, ECG, etc) and process these data mentally to arrive at a diagnosis. This problem will be further compounded by the availability of large amounts of data on genomics, and from wearables that provide continuous measures of heart rate, activity and other indicators of health. Digital twins are a set of mathematical models that use all these different types of data 'inputs' to make predictions, such as the underlying diagnosis of the person from whom the data were obtained. Because digital twins are updated regularly or continuously with new data, they can be used over time to monitor health and disease.[25] Digital twins are not an 'intervention', but instead act as a decision support tool to the patient and clinician to inform decisions about diagnosis, treatment or future risk of disease or deterioration.[24 22] Their potential benefit would be realised if their use led to more accurate or earlier diagnosis of CVD leading to improved patient outcomes and greater healthcare efficiency. However, digital twins remain the subject of research and have yet to translate into clinical care pathways,[25–27] meaning that a greater understanding of the barriers to diagnosis of CVD may influence the development and implementation of this and other digital technologies.

A recent review of telehealth use during the COVID-19 pandemic outlined lack of human contact in care,

confidentiality, data security, and accessibility and training in the use of new platforms as key challenges associated with the implementation of technology into healthcare.[20] Healthcare technologies need to consider how these challenges impact patient and clinician engagement and more work is needed to improve our understanding of user experience to produce sustainable improvements in CVD diagnosis which can be readily implemented into clinical care. Existing work has shown the critical requirements of a digital technology platform for the self-management of CVD[28]; however, the point of self-management can only be reached once an accurate and timely diagnosis has been made.

Digital technologies could play a greater role in facilitating CVD diagnosis, but this requires a deeper understanding of stakeholder experiences of CVD diagnosis, to support development of technological solutions which meet these needs. Given the lack of existing evidence about the lived experience of heart disease diagnosis, we took a qualitative approach to investigate the following objectives:

1. Understand the range of challenges faced by stakeholders in diagnosis of CVD.
2. Understand potential discrepancies between patient and clinician experiences of CVD diagnosis.
3. Make recommendations for requirements of future health technology solutions for improving CVD diagnosis.

## METHODS AND MATERIALS

Our protocol detailing the methodology and procedure has been published.[29] The study was conducted and reported according to Consolidated criteria for Reporting Qualitative research[30] guidelines. The question topic guide involved two main parts—clinical experiences and technology-related experiences—however, the present study includes data related to stakeholder perspectives on contributors to delayed and inaccurate diagnosis of CVD.

### Study design

A qualitative approach was taken to capture the depth of experiences and the complex nature of living with a CVD, as this would not be easily achieved using quantitative approaches. We used semistructured focus groups with people living with CVD to generate discussions of shared experiences during their diagnosis journey, and to allow for direct comparisons between a range of diverse medical experiences which may have been missed or different to information that was collected in a one-to-one interview.

We also conducted one-to-one interviews with clinicians to increase our flexibility around their schedules and collect information across a range of different clinical specialties.

### Patient and public involvement

All participant-facing materials were reviewed by a Sheffield-based cardiovascular patient advisory group. This ensured the information sheet, consent form and

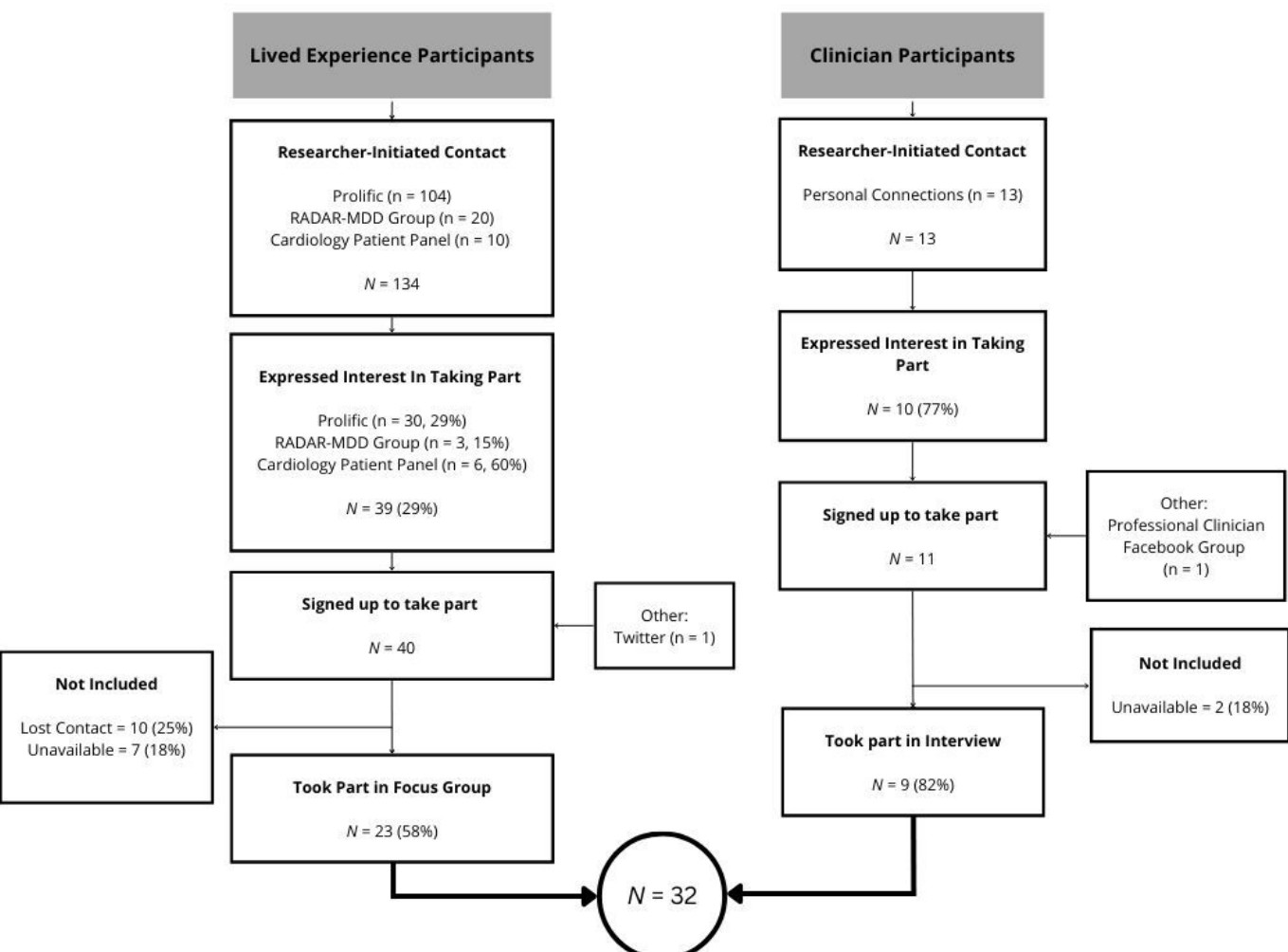

**Figure 1** Breakdown of participant recruitment. RADAR-MDD, Remote Assessment of Disease and Relapse in Major Depressive Disorder.

focus group topic guide were accessible and easy to understand.

## Study population

Figure 1 shows a flow chart of participant recruitment. Inclusion criteria for individuals with lived experience were: a previous diagnosis of CVD, aged 18 years or over, able to speak English sufficiently for participation and able to consent to participate. Exclusion criteria included: major cognitive impairment or dementia preventing participation. The inclusion criteria for clinicians were: >6 months of experience in diagnosis of CVD, aged ≥18 years, able to speak English and able to consent to participation.

The number of participants recruited for focus groups and interviews was based on the available time for data collection against the wider project deadlines and the research team's previous experience conducting qualitative research with clinicians.[29] A maximum of six participants were included in each of the four focus groups to allow adequate time for each participant to share their views and experiences.

## Procedure

Participants with lived experience were recruited using convenience sampling via: Prolific, a cardiology patient panel; and participants from the Remote Assessment of Disease and Relapse in Major Depressive Disorder research study who had consented to be contacted for future research purposes.[31] Study details were also shared on Twitter. Individuals interested in participating were contacted via email to arrange an introductory phone call to confirm interest and eligibility. In this meeting, FM described the research and the procedure of the study.

Clinicians were recruited using purposive sampling via personal and professional connections, and a registered general practitioner (GP) Facebook group. Between them, clinicians represent a range of coverage across the CVD pathway, from diagnosis through to long-term management. However, for the purposes of this study, we exclusively recruited those who diagnose possible CVD on a regular basis. All information was given to clinicians via email prior to the online interview.

Consent and baseline demographic data were collected via online Qualtrics surveys prior to qualitative data

collection (online supplemental file 1). The focus groups and interviews follow a pre-approved, semistructured question schedule, split into two sections (online supplemental file 2). Each focus group included either five or six participants. All focus groups and interviews were conducted online using Zoom (https://zoom.us), with focus groups lasting about 90 min and interviews ranging between 30 and 90 min, based on clinician availability. Interviews and focus groups were facilitated by KA, a female psychology graduate, working full-time on the project. KA had no ongoing relationship with the participants and was not involved in their clinical care. She had neither prior experience in cardiology nor assumptions or expectations of the data. Data collection was supported by two research facilitators (OG and AS) who made field notes during focus groups. Field notes were destroyed once transcripts were deidentified and finalised. Participants were compensated for their time with a £25 Amazon voucher.

The focus group and interviews were audio-recorded, anonymised then transcribed verbatim prior to analysis. Transcripts were validated by KA, OG and AS to confirm transcript accuracy.

### Data analysis
Data relating to patient and clinician experiences of the CVD diagnosis pathway were included in the current analysis. Sample sociodemographic characteristics were described, alongside Depression, Anxiety and Stress Scale[32] scores to understand underlying levels of depression, anxiety and stress at the time of participation. Common mental disorders often exist alongside CVDs[33–35] so it is useful to understand the prevalence of these in our participants. Overall scores were classified into three severity groups—normal (subclinical), moderate and severe—based on validated thresholds.[36]

We conducted an inductive thematic analysis using a phenomenological approach, as this allowed us to be led by the data when exploring emerging themes related to stakeholder experiences. Our method was characteristic of a small q approach, as we followed the postpositivist framework of qualitative analysis to ensure reliability of the resulting themes related to stakeholder experiences of CVD diagnosis.[37] KA used NVivo to conduct the first round of analysis, following the steps recommended by Braun and Clarke.[38] This involved first becoming familiar with the data, followed by an initial code generation and theme identification. After this initial round of analysis, secondary coding and review were conducted by OG and AS to validate theme extraction and support in the naming of themes. The findings were discussed between each of the reviewers before a final decision was made regarding themes and subthemes to be reported.

For the last stage of analysis, we collaborated with an experienced patient and public contributor with experience of conducting qualitative research (LL) to create an additional layer of validation for our framework. This involved meeting over Zoom to discuss our coding framework and reinterpreting the data to accurately reflect a patient perspective. Although data saturation was not assessed, the lived experience insight provided by LL gave us confidence that our data provided both adequate depth and breadth of stakeholder experiences.

### Scientific rigour
To increase scientific rigour of our findings, the results of the first round of thematic analysis were presented to clinicians in the form of a research poster at the British Cardiology Society conference to increase transferability of our results to a wider sample. A QR code was provided next to the poster allowing clinicians to scan it and provide their reflections on whether we captured their experiences or comment on what was missing. Those unable to scan the code (eg, did not have a mobile available on hand) provided verbal feedback to the research poster presenter (KA). Feedback from five clinicians was integrated into later stages of analysis.

We also consulted with the London-based National Institute for Health and Care Research Maudsley Biomedical Research Centre's Race, Ethnicity and Diversity advisory group to provide further cultural insight into our preliminary findings, which were presented via a series of presentation slides summarising the key findings so far. Verbal discussions were facilitated by the lead researcher (KA) and written up to be discussed within the research team and incorporated within later stages of analysis.

Neither form of cross-validation resulted in major changes to the analysis; however, it supported the organisation and description of the themes and subthemes reported. While it is not possible to remove the subjective bias of the researchers conducting the analysis, this patient and public involvement (PPI)-led approach to thematic analysis increases the credibility of our findings, which ultimately increases its translatability beyond our sample.

### RESULTS
### Sample demographics
Four focus groups (N=23) and nine interviews were conducted with a total of 32 individuals contributing data to the study. This represents 63% of interested individuals and 22% of individuals initially contacted. Table 1 summarises the demographic and clinical characteristics of the sample.

Four overarching themes and 34 subthemes were identified (figure 2) and quotes for each subtheme are presented in online supplemental file 3. The four major themes were: *symptom interpretation*, *patient characteristics*, *patient–clinician interactions* and *systemic challenges*.

### Theme 1: symptom interpretation
Our findings revealed the wide variety of symptoms experienced prior to diagnosis—from 'sweating' and 'tachycardia' to 'swollen legs' and 'severe acid reflux'— highlighting the diversity in experiences between patients

**Table 1** Demographics of the sample (N=32)

| | | Total sample (N=32) | Lived experience (N=23) | Clinicians (N=9) |
|---|---|---|---|---|
| Age (years), M±SD (range) | | 58.0±12.2 (31–76) | 61.3±11.5 (31–76) | 48.5±9.1 (35–60) |
| Gender, N (%) | Male | 22 (68.8) | 16 (69.6) | 6 (66.7) |
| | Female | 10 (31.3) | 7 (30.4) | 3 (33.3) |
| Ethnicity, N (%) | White | 27 (84.4) | 21 (91.3) | 6 (66.7) |
| | Asian | 4 (12.5) | 2 (8.7) | 2 (22.2) |
| | Black | 0 (0.0) | 0 (0.0) | 0 (0.0) |
| | Other (Arab) | 1 (3.1) | 0 (0.0) | 1 (11.1) |
| Income bracket, N (%) | Less than £15 000 | 6 (18.8) | 6 (26.1) | 0 (0.0) |
| | £15 000–24 000 | 4 (12.5) | 4 (17.4) | 0 (0.0) |
| | £24 000–40 000 | 8 (25.0) | 7 (30.4) | 1 (11.1) |
| | £40 000–55 000 | 5 (15.6) | 5 (21.7) | 0 (0.0) |
| | More than £55 000 | 7 (21.9) | 1 (4.3) | 6 (66.7) |
| | Not disclosed | 2 (6.3) | 0 (0.0) | 2 (22.2) |
| DASS scores, N (%) | Normal (subclinical) | 23 (71.9) | 15 (65.2) | 8 (88.9) |
| | Moderate | 5 (15.6) | 4 (17.4) | 1 (11.1) |
| | Severe | 4 (12.5) | 4 (17.4) | 0 (0.0) |
| Years of service | 5–10 | – | – | 1 (11.1) |
| | 10–15 | – | – | 0 (0.0) |
| | 15–20 | – | – | 2 (22.2) |
| | 20+ | – | – | 6 (66.7) |
| Clinician role | Primary care | 4 | – | 4 |
| | Secondary care | 4 | – | 4 |
| | Emergency | 1 | – | 1 |
| Heart disease diagnosis | Myocardial infarction | – | 12 (52.2) | – |
| | Heart valve disease | – | 4 (17.4) | – |
| | Atrial fibrillation | – | 6 (26.1) | – |
| | Pacemaker or defibrillator implanted | – | 4 (17.4) | – |
| | Cardiac arrest | | 3 (13.0) | |
| | Angina | – | 6 (26.1) | – |
| | Other* | – | 8 (34.7) | – |
| Non-cardiac comorbidities | Diabetes | – | 6 (26.1) | – |
| | Depression | – | 4 (17.4) | – |
| | Stomach/digestive disorder | – | 5 (21.7) | – |
| | High blood pressure | – | 6 (26.1) | – |
| | Other chest trouble | – | 3 (13.0) | – |
| | Back trouble | – | 3 (13.0) | – |
| | Musculoskeletal disorder | – | 3 (13.0) | – |
| | Autoimmune disease | – | 2 (8.7) | – |
| | Other† | – | 14 (60.9) | – |
| | None of the above | – | 8 (34.8) | – |

DASS clinical threshold scores are as follows: ≤6: normal; 7–12: moderate; >12 severe levels of distress.
Autoimmune disease: rheumatoid arthritis and psoriasis.
Musculoskeletal disorders: knee replacement and osteoarthritis.
*Other heart disease diagnoses include: heart rhythm disorder, stent, left bundle branch block, atherosclerosis, atrial flutter, aortic aneurysm, hypertrophic cardiomyopathy.
†Other comorbidities include: asthma, cancer, ADHD and anxiety, fibromyalgia, migraine, mobility issues, pineal cyst and central serous retinomyopathy, psoriasis, crushed foot, bilateral knee replacement, stroke, osteoarthritis, kidney trouble. Diagnostic groups are not mutually exclusive, participants were able to report having more than one condition.
ADHD, attention deficit hyperactivity disorder; DASS, Depression, Anxiety and Stress Scale.

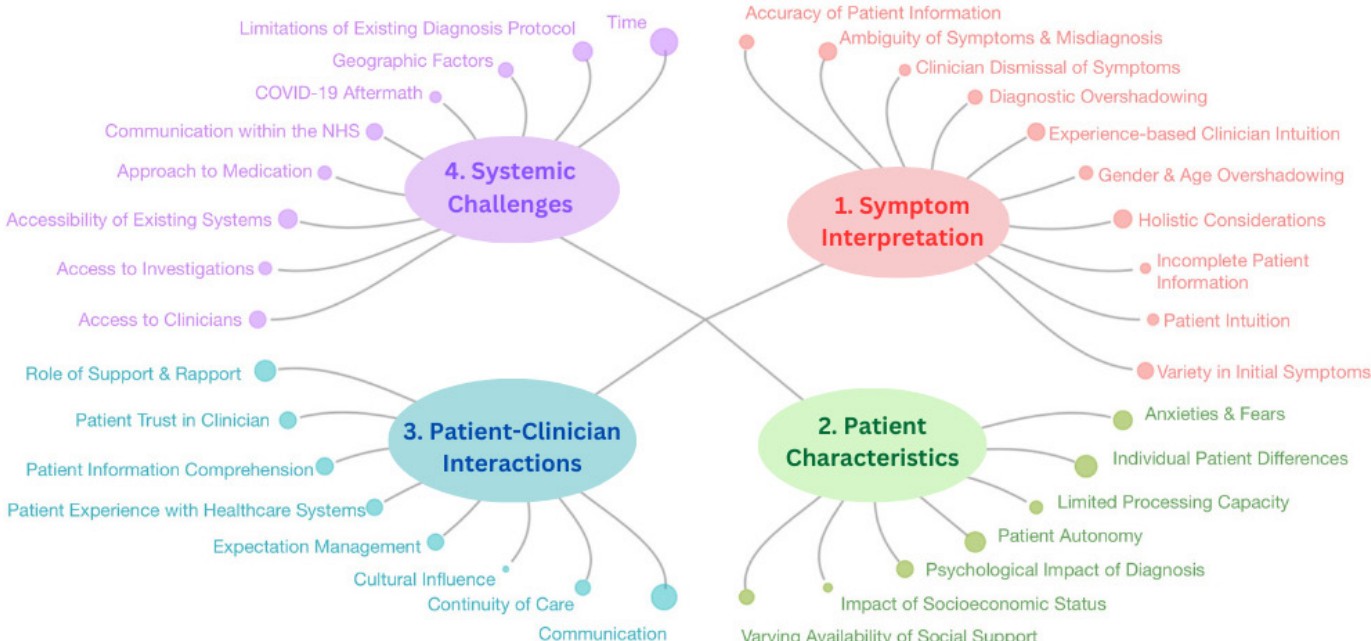

**Figure 2** A breakdown of the four themes and subsequent subthemes emerging from the data. End nodes correspond to the number of codes related to each subtheme; the larger the node, the more instances of coding for that subtheme.

with conditions ranging from atrial fibrillation (AF) to myocardial infarction. From the patient perspective, the ambiguity of symptoms they were experiencing added to confusion about whether to reach out to their doctor: 'is this AF or am I just getting myself worked up with it? So, you kind of doubt your reliability, don't you?' (P18).

Difficulty interpreting the cause or severity of their symptoms negatively affected patients' ability to express themselves to a clinician: 'it really concerns me then, when I'm sat in front of the person who needs to know what has been happening that I can't articulate it very accurately' (P17). Similarly, clinicians describe how patient interpretation of their own condition can affect their ability to efficiently diagnose them and how their personal biases affect the information patients choose to share, for example, when it comes to alcohol, there is a 'natural will to reduce down the amount that they're saying that they're actually drinking' (CL10). This can result in incomplete information being passed on to the clinician, preventing informed decisions regarding their diagnosis. Although clinician perspectives seemed to prioritise 'looking at patient background, and their socio-economic background, their personal family history… smoking as well, diet, stress, exercise, basically the whole picture' (CL12), patients had mixed experiences when it came to holistic considerations of their health condition: 'their job seems to be more physical in terms of, you know, treating the condition rather than, you know, the mental aspect of it or the ongoing aspects of it' (P29).

Patients also described challenges related to diagnostic overshadowing: 'whenever I went to my GP and said, I'm not feeling well, he would say, well, you have a heart problem but I'm pretty sure that the last 2 years I've been suffering from something other than my heart, or

in addition to my heart' (P9). This problem was echoed by clinicians, who acknowledged that 'chest cardiac symptoms can sometimes be very vague and overlap with other diagnoses' (CL9), contributing to false attribution of non-cardiac symptoms to a CVD, or vice versa. Demographic factors, such as age and gender, also created difficulties with symptom interpretation: 'One of the consultants told me that I don't look like a heart problem because I am still young, and I don't, I'm not overweight, or I don't drink or alcohol anything like that. So, I shouldn't have a heart problem basically' (P13), highlighting how members of groups less likely to suffer from heart disease might have less attention paid to their symptoms and experiences.

Both clinicians and patients described experiences related to intuition and previous experience guiding decisions regarding their symptom interpretation, with one patient sharing how they 'just woke up and knew something wasn't right and when the ambulance came and they said, what's wrong? And I said, I don't know but something is' (P20) and a clinician describing how they depend on their 'clinical knowledge' (CL1) to help them interpret a patient's history, in addition to 'biology, physiology, pharmacology, pathology and histopathology'.

This theme provides insight into the important role that digital technologies could play supporting patients and clinicians to interpret systemic and overlapping symptoms linked to possible underlying CVD.

### Theme 2: patient characteristics

Focus group discussions revealed a vast array of individual differences among patients which influence their personal experience of being diagnosed with CVD, with one patient nicely summarising how 'everyone is totally

different, as is what they want, what they want to know, how much they want to know and how they want to know it' (P4). These differences manifested in several ways, for example, in the level of autonomy patients adopted when providing information regarding their health, with one individual describing how their GP stopped 'following up, asking for my readings, so I just stopped doing it' (P10), while another patient 'just took myself to the GP' (P28). Meanwhile, clinicians seem to expect quite a high level of autonomy from patients, as 'there's no active monitoring. The monitoring itself would depend on the patient staying in touch with me' (CL12). There were differences in how much support patients had during their diagnosis with some reflecting how 'they owe a lot of people a lot' (P10) for their support, while others said their 'greatest challenge was I live on my own… and there was no support' (P9). Clinicians also discussed how differences in socioeconomic status will impact diagnosis and patient care, as lower literacy levels require more support, with one clinician describing how they 'bring them in, and I've got forms and sheets that I go through with them and give them information' (CL10).

There was also substantial evidence for the psychological impact of CVD diagnosis, with one patient describing how their 'life seemed to stand still. Not just physically, but also mentally' (P5,) and how there was 'a lot of anxiety around the condition, particularly whilst or once I was diagnosed…that was getting really quite troublesome' (P17). Similarly, clinicians brought up similar issues regarding fear and anxiety management, reflecting on the importance of 'trying to remove the fear from the patient trying to de-escalate them' (CL12) and considering patient mental processing capacity during the time of diagnosis and how this might lead patients to 'delete… and distort a lot of the information' (CL10).

This theme provides insight into how healthcare services and procedures could improve quality of care and patient outcomes by considering patients as individuals, with different needs and expectations.

### Theme 3: patient–clinician factors
Focus group discussions revealed frustrations from some patients regarding the lack of communication and expectation management—leaving them 'initially a bit confused' (P25) and unsure of 'what the future was going to hold or how long I was going to survive' (P29) —and the lack of continuity of care, as 'now if you… phone to speak to a doctor, you invariably get somebody you've never spoken to before' (P22). Clinicians expressed how time affects communication. They are 'only given 10/15 min to see one patient, which I don't think is enough' (CL9) and awareness of lack of continuity of care means that 'particularly in our older clientele…they feel abandoned' (CL10).

Meanwhile, other patients relayed the opposite sentiment, providing positive stories of how 'the cardiologist… certainly did a lot to save my family from me dying' (P2) and a general sentiment of trust in clinician competence:

'you get to realize that they know 100 percent more than you do so it's common sense to accept what they are saying' (P26). Clinicians also described techniques they use to help patients understand their diagnosis, 'like drawings, pictures, things that would be able… to inform the patient better about the condition' (CL7).

There was an element of luck in the type of experience patients had, with some patients describing how lucky they were to have been 'seen by a very astute newly qualified GP' (P28), while others felt their doctors 'were guessing much of the time' (P9). Clinicians seemed to be aware of this, stating that 'depending on the consultant they get, they will feel very informed and very supported, or they will equally feel very judged and dismissed' (CL10). Many of the challenges related to patient–clinician interactions were described by clinicians as being out of the hands of GPs and cardiologists, as they were the result of systemic protocols and not individual decisions, for example, one clinician recalls how she does not 'start drugs going into a bank holiday, because I know the out of hours doesn't exist' (CL10). Nonetheless, interview data suggested clinicians played an active role in mitigating the impact of existing systemic limitations and creating a relationship based on trust and open communication, as they emphasise that 'the patient needs…to be willing to work with the clinicians and if the patient refuses to work with the clinician then no matter how hard the clinicians try you know it's not going to work' (CL7).

This theme provides insight into the importance of building meaningful relationships between patients and clinicians for improving patient outcomes and indicates areas where greater standardisation of care may be of benefit.

### Theme 4: systemic challenges
Many systemic issues affected both patients and clinicians, such as 'the complete lack of communication between the medical staff, the nurses in this case and the senior doctors and…surgeon' (P19) and the fact that 'ED has no mechanism to give [GPs] feedback' (CL10). Both groups also faced obstacles resulting from existing processes, such patients having to go 'through a kind of interrogation to find out how serious' (P8) their condition was to be able to book an appointment, or clinicians who 'feel very strongly that this is a ischaemic heart disease…but if it doesn't meet the criteria, and then it is rejected' by specialists to whom the patient has been referred (CL12). In addition to the limitations of existing protocols, there are several resource-related challenges that are preventing patients from receiving an accurate and efficient diagnosis, including access to clinicians, access to investigations and time-related difficulties. Both patients and clinicians describe difficulties getting appointments, with one patient having 'only seen the cardiologist once in 2 years' (P9), while one clinician boldly stating that 'we need to empower a patient as much as possible… because the NHS [National Health Service]… can no longer provide that sort of mollycoddling' (CL8). However,

**Table 2** Breakdown of theme and subtheme size

| Title | Lived experience group total | Clinician group total | Sample total |
|---|---|---|---|
| Symptom interpretation | 52 | 79 | 131 |
| Holistic considerations | 7 | 13 | 20 |
| Ambiguity of symptoms and misdiagnosis | 15 | 4 | 19 |
| Experience-based clinician intuition | 0 | 18 | 18 |
| Variety in initial symptoms | 16 | 0 | 16 |
| Accuracy of patient information | 5 | 7 | 12 |
| Diagnostic overshadowing | 9 | 3 | 12 |
| Gender and age overshadowing | 9 | 2 | 11 |
| Clinician dismissal of symptoms | 8 | 0 | 8 |
| Patient intuition | 8 | 0 | 8 |
| Incomplete patient information | 2 | 5 | 7 |
| Patient characteristics | 56 | 69 | 125 |
| Individual patient differences | 11 | 20 | 31 |
| Patient autonomy | 12 | 14 | 26 |
| Anxieties and fears | 16 | 6 | 22 |
| Psychological impact of diagnosis | 10 | 7 | 17 |
| Varying availability of social support | 13 | 0 | 13 |
| Limited processing capacity | 7 | 3 | 10 |
| Impact of socioeconomic status | 0 | 6 | 6 |
| Patient–clinician interactions | 78 | 80 | 158 |
| Communication | 24 | 20 | 44 |
| Role of support and rapport | 12 | 16 | 28 |
| Patient information comprehension | 8 | 10 | 18 |
| Patient trust in clinician | 10 | 7 | 17 |
| Expectation management | 5 | 11 | 16 |
| Patient experience with healthcare systems | 14 | 2 | 16 |
| Continuity of care | 7 | 7 | 14 |
| Cultural influence | 0 | 5 | 5 |
| Systemic challenges | 90 | 81 | 171 |
| Time | 23 | 28 | 51 |
| Efficiency | 13 | 1 | 14 |
| Time restrictions | 1 | 14 | 15 |
| Waiting times and delays | 9 | 13 | 22 |
| Limitations of existing diagnosis protocol | 15 | 8 | 23 |
| Accessibility of existing systems | 8 | 14 | 22 |

Continued

**Table 2** Continued

| Title | Lived experience group total | Clinician group total | Sample total |
|---|---|---|---|
| Access to clinicians | 8 | 9 | 17 |
| Communication within the NHS | 5 | 11 | 16 |
| Geographical factors | 7 | 6 | 13 |
| Approach to medication | 11 | 0 | 11 |
| Access to investigations | 0 | 10 | 10 |
| COVID-19 aftermath | 4 | 4 | 8 |

NHS, National Health Service.

other clinicians did not share this sentiment and felt that 'the poor patient is stuck in the middle' (CL2) of issues related to the NHS and even though 'what is lifesaving usually gets done…for life changing procedures…there is no capacity to see all these people so quickly' (CL1).

Geographical factors appeared to exacerbate these issues, as in isolated regions 'the nearest cardiologist is… over several hundred miles away' (CL9) and there is 'a real practical issue about getting access to tests' (CL8). There is some evidence to suggest that patients and clinicians feel previously existing systemic issues, particularly in relation to access to clinicians, has worsened as a result of the COVID-19 pandemic, as patients feel 'you can't get to see [the GP] obviously as easily' (P18) and clinicians are also aware that 'in the kind of post-Covid era, patients have a lot of problems getting access to a GP so I mean, that's a big barrier right now, and we have we have a limited resource' (CL13).

Finally, there were also patient concerns with the approach to medication within the healthcare system and a general lack of understanding in the way their medications were managed (or were not managed) by their clinicians, as 'no one questions' repeat prescriptions (P5) or keeps track of the dosage until the patient questions it. Overall, our data showed how existing issues within the healthcare system interact with each other to ultimately disadvantage both patients and clinicians, and as a result, contribute to poorer outcomes for patients with CVD.

This theme highlights how limited access to resources, in the form of clinicians, investigations and time, acts as barriers to efficient and accurate diagnosis of CVD. The better use of digital technologies to obtain, represent and allow transfer and sharing of an individual's information between different teams of healthcare professionals could address some of these issues.

Table 2 breaks down the size of each theme and subtheme, split by population group. Despite variation in subtheme sizes between the groups, the overall totals for each major theme are similar, suggesting that patients and clinicians face comparable challenges, despite some variation in more specific difficulties captured by the subthemes.

## DISCUSSION

Our study suggests clinicians and patients face a variety of challenges preventing accurate and timely diagnosis of CVD. These difficulties were categorised by experiences related to symptom interpretation, patient characteristics, patient–clinician interactions and systemic challenges. These four major themes were relatively similar in size, although the systemic challenges theme was largest and the patient characteristics theme smallest.

Challenges related to 'time', including time restrictions, long waiting times and delays, and efficiency, had the greatest number of references, consistent with known issues with time and resources within the NHS.[39] The second biggest subtheme was 'communication' between patients and clinicians, including a combination of positive and negative patient experiences. This is consistent with a previous qualitative study including both patients and healthcare professionals, which highlighted how problems with communication lead to a lack of patient understanding of their CVD.[40] A systematic review found that patient–clinician interactions influence patient capacity to engage in self-care for their heart condition via their ability to influence patient understanding of their condition.[41] Thus, our study complements wider knowledge of the direct and indirect influences of quality communication on patient healthcare outcomes.[42 43] The third largest subtheme was related to 'individual patient differences', highlighting the role of patient differences in determining both patient and clinician experiences during diagnosis. Although current healthcare has significant limitations, it does at least have the theoretical capacity for the clinician to tailor some (but not all) aspects of the medical encounter to the needs, understanding and preferences of the patient. However, it is clear that digital tools without the capability to be customised to the individual user (either patient or clinician) run the risk of failing to adapt to these patient differences. It is important for such digital technologies to be designed to be adaptable enough to account for such differences. Conversely, the ability to provide data where and when the patient feels most comfortable at their own pace, and to do so outside of a potentially stressful medical encounter, may provide opportunities to account for these differences. Ultimately, this could improve patient experience and health outcomes in ways that would not be possible in the traditional healthcare pathway which has fixed times and locations in which data are obtained.[44 45]

There were several smaller subthemes found in our data that are supported by existing literature. For example, we found that location contributed to systemic challenges such as access to clinicians and investigations, consistent with previous studies investigating the variability in access to diagnostic tests[46] and inequity in GP supply[47] across the UK. Also, previous findings related to post-COVID-19 challenges in CVD diagnosis were supported by both patient and clinician perspectives, although the smaller size of the subtheme may be due to many of our patient experiences being pre-COVID-19 and thus were not affected by delays following nationwide lockdowns.[18 48] However, issues related to diagnostic overshadowing and misdiagnosis due to comorbidities were less substantial than expected, given existing knowledge of difficulties diagnosing vague symptoms and falsely attributing overlapping symptoms to pre-existing non-cardiac conditions.[49 50] This may be due to clinicians not feeling comfortable admitting that they struggle to accurately diagnose their patients; meanwhile, 'ambiguity of symptoms and misdiagnosis' came out as one of the largest subthemes from patient data, suggesting that difficulties with accurate symptom interpretation can prevent some patients from reaching out to healthcare professionals.

Our results shed light on the differences between patient and clinician experiences, highlighting the importance of considering how barriers to diagnosis may be affecting each group differently. The weight of evidence for the four major themes did not differ substantially between lived experience and clinician groups; however, there were differences between the size of subthemes. Although 'time' and 'communication' were the largest subthemes for both participant groups, 'individual patient differences', 'role of support and rapport' and 'experience-based clinician intuition' were the largest subthemes from clinician data, while 'anxieties and fears', 'ambiguity of symptoms and misdiagnosis' and 'limitations of existing diagnosis protocol' were the largest subthemes from patient data. Notably, both participant groups acknowledged in some way the psychological aspects of CVD diagnosis, which is supported by existing findings showing moderate levels of depression and anxiety and significant life changes in people with heart failure.[51] Nevertheless, patient data revealed more about the patient mental load associated with diagnosis, while clinicians shared their opinions and experiences related to providing support for patients, which is in line with previous research looking at the role of clinicians in providing relief and support following a cardiac event.[52]

A summary of considerations for digital twin technologies arising from this research is listed in table 3. These could also be used for future research and development of other health technology solutions aiming to improve accuracy and efficiency of CVD diagnoses and improve patient outcomes by reducing mortality and increasing treatment efficacy.

### Strengths and limitations

A key strength of the present study is the use of qualitative interviews and focus groups to achieve its objectives, as this allowed for open-ended questions to provide a more in-depth understanding of the current challenges faced by stakeholders in diagnosis for CVD. Specifically, the decision to carry out focus groups with lived experience groups allowed for lively discussions where participants felt more comfortable to disclose their personal experiences and relate to their peers.

Our inclusion of both patient and clinician experiences allowed a more complete and integrated understanding

**Table 3** Considerations for future research

| Considerations | Potential design aspects of digital twin technologies |
|---|---|
| Improve efficiency of processes and interactions during diagnosis, to aid both clinicians and patients when time is limited | Allow capture of symptoms and other aspects outside of consultation and present these to patient and clinician in a meaningful way to reduce the time needed to obtain such information in clinic |
| Implement strategies to improve communication between patients and clinicians so patients feel informed throughout the process of their diagnosis and have access to accurate and appropriate information | Provide feedback and information to the patient about their symptoms and clinical pathway, rather than a one-way flow of information from the patient into the healthcare system |
| Acknowledge individual patient differences when implementing any solutions or new systems by allowing personalisation and flexibility in the use of the solution | Provide customisable innovations that can be personalised/adapted as required by the user |
| Incorporate support and rapport building between patients and clinicians in systems aimed to improve accuracy and efficiency | Augment and improve, not replace personal interactions, between patient and clinician |
| Be conscious of the extent to which patients are expected to take autonomy over their care and how (in)capable they are of managing their symptoms independently without support from clinicians and healthcare services | Monitor and reduce the 'work' required by patients to complete questionnaires, wear devices, etc so that the burden of responsibility is not unfairly shifted from healthcare system to patient |
| Make future solutions more inclusive for patients of different ages, literacy levels, mental and physical health conditions. | Maximise accessibility via co-design and rigorous testing in diverse groups |
| Include ways to manage the psychological impact of diagnosis and the indirect effects on quality of life | Measure psychological impacts during the diagnostic process using approaches such as validated questionnaires and provide interventions to mitigate these impacts when recognised |
| Acknowledge the wider context of the individual beyond their symptoms to encourage a holistic approach to diagnosing CVD | |
| Reflect on the wide variety of typical and non-typical symptoms related to CVD and provide better information for patients to help improve their ability to recognise when it is appropriate to reach out to health services | Ensure digital technologies capture a broad range of symptoms and experiences rather than very limited aspects (such as a focus on chest pain or breathlessness) |
| CVD, cardiovascular disease. | |

of the barriers preventing accurate and efficient CVD diagnosis. Most existing studies assessed these two groups separately[46 53–56] limiting opportunity to see how needs and requirements compare. A decentralised recruitment strategy also meant the participant sample included a range of individuals from across the country, with a variety of CVD diagnoses.[57] Although the majority of the clinician sample was highly experienced, this may have introduced a bias in their perspectives related to barriers, for example, their years of experience may have made them more or less affected by certain issues. Future studies should ensure a variety of levels of clinical experiences are considered to prevent potentially biased interpretations. Nonetheless, the contribution of PPI groups to the design, recruitment and analysis process, and cross-validation of preliminary findings with a range of clinicians at the British Cardiovascular Society Conference increases the transferability of our findings.

Our sample suffered a lack of ethnic diversity, particularly in our patient group. This may explain why we did not find patient data to support the role of 'cultural influence' on patient–clinician interactions. We attempted to remedy this limitation by consulting with a Race and Ethnicity advisory group, who suggested that we might be missing data on culturally specific patient experiences related to family and religion among ethnic minorities. Stratified sampling may facilitate adequate ethnic diversity and representation in future research studies.

The use of online recruitment platforms and snowball convenience sampling to recruit our participants may have produced a biased sample of individuals who are more involved in their own healthcare and new technological developments in cardiovascular area. Therefore, our sample may be less representative of patient and clinician populations who are less digitally literate who may face even greater challenges in receiving or delivering accurate and efficient CVD diagnosis. Future research should consider ways to include more seldom-heard groups in research investigating contributors to delayed and inaccurate diagnosis of CVD.

Finally, we did not collect data on when participants were diagnosed with their CVD. This information could have been useful to understand how lived experiences varied for participants who were diagnosed more recently compared with those who were diagnosed decades ago. There also could be greater recollection bias from participants who were describing experiences from a long time ago, which undermines the quality of evidence. Future studies exploring clinical experiences of diagnoses could specify a cut-off date during recruitment to avoid this potential bias in the data.

Moreover, further investigation could be done to determine whether the present study's findings are consistent with patients currently undergoing diagnosis, especially since the COVID-19 pandemic, as this may highlight the most urgent areas that would benefit from novel digital health technologies.

There were emerging trends of gender-specific experiences from our patient group that require further investigation, especially given the growing literature exposing how women are at a greater disadvantage when it comes to receiving an accurate and timely CVD diagnosis.[54 56 58 59]

## Implications for digital technology development

Several considerations have been suggested that would inform development of digital twin and other technological innovations to improve the accuracy and efficiency of CVD diagnosis. Such technologies must overcome key barriers related to time, patient–clinician communication and difficulties tailoring to individual patient differences within the diagnosis pathway. Successful innovations need to increase efficiency, improve patient–clinician communication and to provide a tailored approach to diagnosing individuals with heart disease. While this study provides insight into patient and clinician experiences of these challenges, further research is required to enhance our understanding of how these experiences differ between ethnic groups and genders.

Although our work focused on diagnosis as the first essential step in a clinical pathway, diagnosis alone does not improve patient outcomes. This requires interventions (such as behaviour change, drug treatment or surgery) and most CVDs also require some form of ongoing monitoring to detect changes over time and attempt to predict deterioration early enough to prevent complications such as hospital admission or death. It is likely that many of the characteristics required for better diagnosis could be transferred to technologies used to guide decisions about interventions and monitoring, but this requires further research.

**Author affiliations**
[1]School of Psychology, University of Sussex, Falmer, UK
[2]Honorary Fellow, College of Health, Wellbeing and Life Sciences, Centre for Applied Health & Social Care Research (CARe), Sheffield Hallam University, Sheffield, UK
[3]Clinical Medicine, School of Medicine and Population Health, The Medical School, The University of Sheffield, Sheffield, UK
[4]Centre for Personalised Medicine, Ulster University Faculty of Life and Health Sciences, Londonderry, UK
[5]School of Computing Sciences, University of East Anglia, Norwich, UK
[6]Imperial College London, London, UK
[7]Department of Mechanical Engineering, University College London, London, UK
[8]Wellcome/EPSRC Centre for Interventional and Surgical Sciences, University College London, London, UK

**Acknowledgements** We would like to thank the two patient and public involvement groups that helped to inform the design of this study: the NIHR Maudsley Biomedical Research Centre's Race, Ethnicity and Diversity (READ) advisory group and the Sheffield-based Cardiology Patient group. We would also like to thank Dr Valerie de Angel for sharing her R code to help us create figure 2 (see online supplemental file 4) and to thank Helen Denney and Amber Ford for convening the Sheffield patient group and for administrative assistance.

**Contributors** Conceptualisation—FM, TC, JC, OB, VD and RVA. Methodology—FM and TC. Investigation—KA, MM, OG, AS and LL. Writing (original draft)—KA. Writing (review and editing)—FM, TC, LL, OG, AS, MM, JC, OB, VD and RVA. Supervision—FM and TC. Project administration—KA and FM. Funding acqusition—FM, TC, JC, OB, VD and RVA. Guarantor: FM

**Funding** This work is supported by the UK Engineering and Physical Sciences Research Council (EPSRC) (grant number: EP/X000257/1).

**Competing interests** None declared.

**Patient and public involvement** Patients and/or the public were involved in the design, or conduct, or reporting, or dissemination plans of this research. Refer to the Methods section for further details.

**Patient consent for publication** Not applicable.

**Ethics approval** This study involves human participants and was reviewed and approved by the Sciences & Technology Cross-School Research Ethics Council at the University of Sussex (reference ER/FM409/1) on 14 November 2022. Participants gave informed consent to participate in the study before taking part.

**Provenance and peer review** Not commissioned; externally peer reviewed.

**Data availability statement** Data are available upon reasonable request.

**ORCID iD**
Kamilla Abdullayev http://orcid.org/0000-0001-6233-5955

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
