## [Reviewer comments · BMJ Open]

ARTICLE DETAILS

TITLE (PROVISIONAL)	STAKEHOLDER PERSPECTIVES ON CONTRIBUTORS TO DELAYED AND INACCURATE DIAGNOSIS OF CARDIOVASCULAR DISEASE AND THEIR IMPLICATIONS FOR DIGITAL HEALTH TECHNOLOGIES: A UK-BASED QUALITATIVE STUDY.
AUTHORS	Abdullayev, Kamilla; Gorvett, Olivia; Sochiera, Anna; Laidlaw, Lynn; Chico, Timothy; Manktelow, Matthew; Buckley, Oliver; Condell, Joan; Van Arkel, Richard; Diaz, Vanessa; Matcham, Faith

VERSION 1 – REVIEW

REVIEWER	Morton, Katherine University of Southampton, Psychology
REVIEW RETURNED	02-Nov-2023

GENERAL COMMENTS	The manuscript describes the findings from interviews with patients and clinicians about their experience of being diagnosed with/diagnosing CVD, with a focus on understanding how best to design digital health interventions to support diagnosis. While the paper is well-written, I've outlined three key concerns below that I think should be addressed: 1. The paper could be clearer about the scope of the clinical context it aims to explore, and the rationale for the study. The introduction mentions 'Digital Twins' as an example of an intervention, but doesn't make it clear whether that is a specific intervention or a type of intervention, or to what extent this study will inform the development of a specific intervention. Also, details about where clinicians were based (Primary/Secondary Care), what their role was, and how much of the patient's CVD journey would be covered would help the reader better understand the context for the study (e.g. some of the data relates to longer-term management of the condition, beyond initial diagnosis, which might not be expected based on the objectives).2. The authors state they used reflexive thematic analysis, but the emphasis on 'validating' the themes, and discussion about the size of each theme to indicate its importance are not consistent with a reflexive TA approach.3. The insights generated felt quite broad and not very unique or specific to digital health interventions. I saw in the topic guide that you had some questions around digital health tools but it doesn't feel like these data have been included in the writeup? Table 3 could be more clearly linked to the findings and perhaps consider more specifically what a Digital Twins intervention could do to support CVD diagnosis, given this is your focus in the introduction?
--

	Intro  - The second para outlines the risks of delayed or misdiagnosis, but it would be interesting to explore the reasons for this. Has any qual or quant research looked at why this occurs? Also the first sentence of the second para is a little hard to follow. - It would be useful to understand more about how Digital Twins work in practice, e.g. what does the patient have to do? What does the clinician have to do? If not yet used in the clinical care pathway, in what context have they been used in the cited references? Methods  - What was the rationale for collecting data on depression, anxiety and stress? - The procedure only mentions recruitment of participants via Prolific and social media, but the flowchart in Fig 1 shows 6 of the patients who expressed interest came from a cardiology patient panel and 3 from another group that isn't mentioned – could you add these into the procedure description? - Analysis: I'm not sure what you mean by saying that you took a 'phenomenological approach' but used reflexive thematic analysis? Also, using second reviewers to 'validate' theme extraction is not compatible with reflexive thematic analysis which emphasises the importance of acknowledging and embracing researcher subjectivity and how this will influence the analysis. If you've used reflexive thematic analysis, it would be good to read and cite a more recent Braun and Clarke paper, e.g. 2019 Reflecting on reflexive thematic analysis and 2023 Is thematic analysis used well in health psychology? A critical review of published research, with recommendations for quality practice and reporting. - Small typo: "We used semi-structured focus groups in people living with CVD" should be changed to 'with people living with CVD" - If saturation was not assessed, how did you determine your sample size? - Scientific rigour: It's not clear how presenting the first round of thematic analysis findings to clinicians as a poster increased the transferability of findings to a wider sample. Could you unpack this a bit more? Did you seek feedback from the clinicians? How was this incorporated into subsequent analysis? Similarly, how did feedback from the READ advisory group and the PPI contributor influence the findings? Results  - I think that Table 1 should sit within the results, not methods. It would be useful to know more about the roles of the clinicians, e.g. were they based in primary or secondary care? And did you collect data on how long it was since a patient's CVD diagnosis? This might be useful to consider when interpreting their recollections of being diagnosed. - How many ppts were in each focus group? - I did not feel very confident in the analysis when reading the findings, as quite often the description and interpretation of a quote did not seem to match the quote itself. Perhaps it would be worth revisiting the description of the themes to consider how the quotes are interpreted? I've given some examples from the first theme below:
--	---

o The quotes on page 11 “is this AF or am I just getting myself worked up with it? So, you kind of doubt your reliability, don't you?” (P18) doesn't in itself show that it was confusing symptoms that gave the patient doubts. It suggests more that they were unsure whether they were overreacting or not.

o The quote on page 12 para 1 (“their job seems to be more physical in terms of, you know, treating the condition rather than, you know, the mental aspect of it or the ongoing aspects of it” (P29)) seems to be about the patient's perspective of whether the clinician takes a holistic approach to treating their condition, not diagnosing it, which is the focus of the research question.

o The second para on page 12 also felt a bit confused to me. The patient quote (“whenever I I went to my GP and said, I'm not feeling well, he would say, well, you have a heart problem but I'm pretty sure that the last two years I've been suffering from something other than my heart, or in addition to my heart”) suggests they were already diagnosed as having a heart problem but felt something else was wrong, so is this about being diagnosed with CVD? The following clinician quote that “chest cardiac symptoms can sometimes be very vague and overlap with other diagnoses” (CL9) is interpreted by the authors as “contributing to false attribution of non-cardiac symptoms to a cardiovascular disease”, but isn't it the other way around? I may have misunderstood this but suggest it needs unpacking a bit more to make it clearer for the reader.

o Para 3 on page 12 is confusing as it aligns a patient's intuition (“knew something wasn't right”) with clinician's holistic approach to applying their knowledge (“knowledge of the biology, physiology, pharmacology, pathology, and histopathology, and as well as clinical knowledge”), which seem very different things.

- Theme 2 quickly moves from discussing diagnosis in the opening sentence, to discussing involvement in managing the condition which is not focus of the research question.

- Table 2 and the frequent references in the following para and throughout the discussion about the 'size' of each theme or 'weight of evidence' feels a very quantitative approach, which again is not compatible with reflexive thematic analysis. Number of mentions of a theme doesn't necessarily indicate how important it is. Perhaps a coding manual with definitions of each code and an example quote would be more useful here, like an abbreviated version of the theme table in your appendix?

Discussion

- Page 20: The suggestion that “The growing interest in the implementation of personalised healthcare via wearable devices and digital medicine provides opportunity to account for these differences and improve patient experience and health outcomes”, feels a sweeping statement, given that the theme around patient characteristics covers diverse patient experiences including anxiety around diagnosis, lack of social support, and clinician perceptions around patients' SES, which digital devices cannot be assumed to address. More nuanced consideration of the possible benefits and issues of using digital devices in this context would be good here.

- The considerations in Table 3 feel very broad, e.g. “Make future solutions more inclusive for patients of different ages, literacy levels, mental and physical health conditions”, is not particularly

	focused on digital interventions in this context nor taking account of the implementation barriers to achieving these.  - Might the study have been strengthened by including some patients with ambiguous symptoms who have not yet been diagnosed, or only including patients who are currently undergoing diagnosis? - Further consideration about the impact of the recruitment approach would be useful to see, e.g. why was twitter used rather than a less academic social network like facebook? Was there any option to sign-up without an email address? From Figure 1 it looks like 1 person was recruited via social media rather than via the existing research groups of CVD patients agreeing to be invited to further research –which might be useful to reflect on? It would also be useful to include the email that was used to contact people initially to ask if they were interested, after they got in touch? Could you also report if people were paid to take part? What steps could have been taken to make it easier for people to take part, e.g. option to do an interview via phone call instead of the Zoom focus group for non-Zoom users, or those who would prefer a 1:1 discussion? You lost 7 participants who were unavailable at the focus group times, but they could have been included had 1:1 interviews been offered. What happened to the 10 people who you lost contact with after they had agreed to take part? - For the clinician interviews, where did the personal connections come from? It was interesting that 6 of the 9 clinicians had completed more than 20 years in the service. How might it have influenced the findings that very few were new to the role? Conclusion The conclusion feels more of a quick wrap-up statement. Perhaps more consideration could be given to what are the really interesting and unique findings from this research in relation to using digital interventions for CVD diagnosis?
--	---

REVIEWER	Kirby, Emma University of New South Wales
REVIEW RETURNED	23-Nov-2023

GENERAL COMMENTS	Thank you for the opportunity to review this study, which aims to better understand delayed and inaccurate diagnosis of cardiovascular disease from perspectives of both patients and clinicians. The study and findings have much to offer, not only in relation to CVD, but in terms of the various subjective positions and perspectives that coalesce to shape the experience of delay in diagnosis. As such, the findings could be most useful in thinking more broadly about how patients/people interpret symptoms, patient-clinician relationships, and how the everyday bureaucracies and challenges within health systems can shape outcomes (indeed, some of these ‘bigger’ implications could, and in my view, should, be engaged with in the Discussion section, as they would further extend the relevance and potential impact of the research). In my view the article could be suitable for publication, once some important methodological and related issues are addressed. I outline these below – I hope that the authors find these suggestions/comments constructive in advancing the article. Essentially, my comments relate to how the study is situated, and how revisions can improve the alignment of aims with approach, analysis, and interpretation of findings.
--

Some explanation of rationale for focus groups and interviews (pragmatics of data collection are mentioned briefly, but some engagement with the appropriateness of these methods for the type of knowledge required to address research aims/questions is also needed). Relatedly – the objectives of the study, as they stand, are not in and of themselves obviously aligned to a qualitative approach – so some justification of approach (relative for example to a survey of patients and clinicians) is needed. As a note – the objectives stated in the previously published protocol have better alignment with a qualitative approach; however, there is also not explicit justification or rationale for this approach (methodologically, epistemologically, theoretically) in the protocol – so the protocol cannot be relied on in isolation to provide the required information.

The main issue in need of addressing for the article to be suitable for publication as a qualitative study is (the current lack of) engagement with a theoretical or conceptual framework. Reflexive thematic analysis as an approach to qualitative analysis requires explicit consideration of theory, and/or explanation of the philosophical, ontological, and epistemological underpinnings of the approach to data and knowledge production. An approach to analysis is not sufficient; an approach to research methodology and an approach to understanding the knowledge that will then be generated is also required. At present, the article is an (albeit very well done) example of more proceduralist approaches to qualitative work – given that Braun and Clarke’s reflexive thematic analysis was drawn on within the study, some inclusion/integration of how a theoretical or conceptual framework guided the research is needed. Braun & Clarke in their more recent work have published on this – engaging with the issues outlined in their below articles would be useful (particularly in terms of rigour of process and approach, as distinct from the issues of scientific rigour currently included in the article).

<https://www.tandfonline.com/doi/full/10.1080/17437199.2022.2161594>
<https://www.tandfonline.com/doi/full/10.1080/26895269.2022.2129597>

Relatedly, the extent of interpretation of the data could be improved. Part of the problem here does not lie with the authors – I want to be clear in acknowledging that journal formatting guidelines, including in BMJ Open, often serve to stifle attempts to show interpretation (e.g., through separation of Results and Discussion sections). That said, some more emphasis on the subjectivities of interpretation of findings in relation to existing theory/concepts – and not just in relation to empirical findings from other studies – would improve the contribution of the article. This will also improve the alignment of the results and discussion with the approach/practices of reflexivity that is an important part of reflexive thematic analysis.

Other minor issues:

- Discussion: how does the size of themes align with an RTA approach? Some consideration of this, and amendment accordingly is needed.

- The emphasis within the study on the creation/development of a digital health tool could be further contextualised, particularly in relation to the thematic findings. Alternatively, the authors might explain in the early parts of the article, why the digital tool and lived experiences of engaging with digital health tools are not the focus of the article.

VERSION 1 – AUTHOR RESPONSE

Reviewer: 1

Dr. Katherine Morton, University of Southampton

Comments to the Author:

The manuscript describes the findings from interviews with patients and clinicians about their experience of being diagnosed with/diagnosing CVD, with a focus on understanding how best to design digital health interventions to support diagnosis. While the paper is well-written, I've outlined three key concerns below that I think should be addressed:

1. The paper could be clearer about the scope of the clinical context it aims to explore, and the rationale for the study. The introduction mentions 'Digital Twins' as an example of an intervention, but doesn't make it clear whether that is a specific intervention or a type of intervention, or to what extent this study will inform the development of a specific intervention.

We apologise for not explaining Digital Twins more clearly. We have expanded this section to explain their potential role not as an intervention per se, but as a decision support tool to guide patient and clinician decision making leading to improved health outcomes and healthcare system efficiency. Please see changes page on page 3:

“A range of potential digital technologies have been proposed to address the challenges in diagnosing CVD (21). One such technology is a “Digital Twin” (22–24). Digital Twins address the problem that healthcare professionals today are required to assimilate huge amounts of data on each patient (including description of symptoms, medical and medication history, lab tests, x-rays and other imaging, physiological measures such as heart rate, blood pressure, ECG etc) and process these data mentally to arrive at a diagnosis. This problem will be further compounded by the availability of large amounts of data on genomics, and from wearables that provide continuous measures of heart rate, activity and other indicators of health. Digital Twins are a set of mathematical models that use all these different types of data “inputs” to make predictions, such as the underlying diagnosis of the person from whom the data was obtained. Because digital twins are updated regularly or continuously with new data, they can be used over time to monitor health and disease (25). Digital Twins are not an “intervention”, but instead act as a decision support tool to the patient and clinician to inform decisions about diagnosis, treatment or future risk of disease or deterioration (24) (22). Their potential benefit would be realised if their use led to more accurate or earlier diagnosis of CVD leading to improved patient outcomes and greater healthcare efficiency. However, digital twins remain the subject of research and have yet to translate into clinical care pathways (25–27), meaning that a greater understanding of the barriers to diagnosis of CVD may influence the development and implementation of this and other digital technologies.”

2. Also, details about where clinicians were based (Primary/Secondary Care), what their role was, and how much of the patient's CVD journey would be covered would help the reader better understand the context for the study (e.g. some of the data relates to longer-term management of the condition, beyond initial diagnosis, which might not be expected based on the objectives).

We agree and have attempted to more clearly explain that digital twins could be used to guide decisions at all points of the CVD journey, but that the data inputs and outputs would differ according to the specific context in which the twin is being applied. We have also included more information about where clinicians are based in the demographic table on page 10 and have changed the text on

page 7 to reflect this:

“Between them, clinicians represent a range of coverage across the CVD pathway, from diagnosis through to long-term management. However, for the purposes of this study, we exclusively recruited those who diagnose possible CVD on a regular basis. All information was given to clinicians via email prior to the online interview.”

3. The authors state they used reflexive thematic analysis, but the emphasis on ‘validating’ the themes, and discussion about the size of each theme to indicate its importance are not consistent with a reflexive TA approach.

Thank you for this insight and we apologise for inappropriately using the term reflexive here. We have revised our understanding of the different approaches to thematic analysis, via the literature recommended by both reviewer 1 and 2, and clarified our description of the methodology to more accurately represent our procedure on page 8. Given the highly clinical readership of BMJ Open, as well as the technology-focused nature of the study, we hoped to provide a more relatable interpretation of the findings while still exploring stakeholder experiences in depth.

We conducted an inductive thematic analysis using a phenomenological approach, as this allowed us to be led by the data when exploring emerging themes related to stakeholder experiences. Our method was characteristic of a small q approach, as we followed the postpositivist framework of qualitative analysis to ensure reliability of the resulting themes related to stakeholder experiences of CVD diagnosis (37). KA used NVivo to conduct the first round of analysis, following the steps recommended by Braun and Clarke (38). This involved first becoming familiar with the data, followed by an initial code generation and theme identification. After this initial round of analysis, secondary coding and review was conducted by OG and AS to validate theme extraction and support in the naming of themes. The findings were discussed between each of the reviewers before a final decision was made regarding themes and sub-themes to be reported.”

4. The insights generated felt quite broad and not very unique or specific to digital health interventions. I saw in the topic guide that you had some questions around digital health tools but it doesn't feel like these data have been included in the writeup?

Thank you for highlighting this – we feel that the breadth of data collected as part of this project is too large to be summarised in one paper. As per our published protocol, we have therefore divided our analysis into separate papers with distinct aims. The aims of the current paper relate to the diagnostic pathway in which the technology might fit; the aims of the second paper will relate to the technological capabilities you've mentioned. We endeavoured to fit all aspects into one paper however, as mentioned by the second reviewer, journal formatting guidelines, including in BMJ Open, often serve to stifle attempts to show interpretation, therefore we will be dedicating a separate paper to the points you've raised here.

5. Table 3 could be more clearly linked to the findings and perhaps consider more specifically what a Digital Twins intervention could do to support CVD diagnosis, given this is your focus in the introduction?

Thank you for your comment. As we have now clarified in the introduction on page 3, we hope it is now clear that the DT is not an intervention. It is a data-driven tool and an AI-based mechanism for supporting clinical decision-making (see response to above comment). We have also added another column to Table 3 which outlines potential design aspects of digital twin technologies which we hope provides a clearer link to our study's findings (see page 23 for the table).

1. Intro

a. The second para outlines the risks of delayed or misdiagnosis, but it would be interesting to explore the reasons for this. Has any qual or quant research looked at why this occurs? Also the first sentence of the second para is a little hard to follow.

Thank you for this insight, we have added information on reasons for delayed or misdiagnosis, as well as adapting the first sentence of the second paragraph of the introduction to make it easier to follow, both on page 3:

“Inequitable access to health care, symptom recognition disparities, structural influences on provision of timely and high-quality care, and bias among clinicians may all contribute to delays in CVD diagnosis (11). There are also likely to be issues related to healthcare access and diagnosis following the COVID-19 pandemic, which led to reduced CVD admissions and increased CVD mortality (12). Delayed or misdiagnosis contribute to inappropriate treatments or unnecessary evaluations (13,14). Adverse outcomes such as hospitalisation or death may result from effective treatment not being received until the disease is more advanced (15). For example, a missed diagnosis of heart failure is associated with increased hospital readmission rates (16), and with a two-fold increased risk of death (17)”.

b. It would be useful to understand more about how Digital Twins work in practice, e.g. what does the patient have to do? What does the clinician have to do? If not yet used in the clinical care pathway, in what context have they been used in the cited references?

Please see the response to the comment above and refer to the changes outlined on page 3.

2. Methods

a. What was the rationale for collecting data on depression, anxiety and stress?

These data were collected with a two fold aims: firstly we needed to assess levels of depression, anxiety and stress to provide suitable onward referral in the case of disclosed emotional distress (as per our ethically approved risk management protocol). Secondly, as CVD is highly comorbid with common mental disorders, we were keen to be able to describe the prevalence of these conditions in our sample for comparison with other cohorts and population level estimates. We have provided a clearer explanation of these reasons for inclusion in the manuscript on page 7:

“Sample sociodemographic characteristics were described, alongside Depression, Anxiety and Stress scores (DASS, (32)) to understand underlying levels of depression, anxiety and stress at the time of participation. Common mental disorders often exist alongside CVDs (33–35) so it is useful to understand the prevalence of these in our participants.”

b. The procedure only mentions recruitment of participants via Prolific and social media, but the flowchart in Fig 1 shows 6 of the patients who expressed interest came from a cardiology patient panel and 3 from another group that isn't mentioned – could you add these into the procedure description?

Thank you for highlighting this inconsistency in terminology, we have adapted the manuscript to ensure consistency on page 6:

“Lived experience participants were recruited using convenience sampling via: Prolific; a cardiology

patient panel; and participants from the RADAR-MDD research study who had consented to be contacted for future research purposes (31).”

c. Analysis: I’m not sure what you mean by saying that you took a ‘phenomenological approach’ but used reflexive thematic analysis? Also, using second reviewers to ‘validate’ theme extraction is not compatible with reflexive thematic analysis which emphasises the importance of acknowledging and embracing researcher subjectivity and how this will influence the analysis. If you’ve used reflexive thematic analysis, it would be good to read and cite a more recent Braun and Clarke paper, e.g. 2019 Reflecting on reflexive thematic analysis and 2023 Is thematic analysis used well in health psychology? A critical review of published research, with recommendations for quality practice and reporting.

Thank you for providing recommendations for appropriate literature. As mentioned in our earlier response, we have adapted the description of our analysis to more accurately reflect our thematic analysis procedure on page 8:

“We conducted an inductive thematic analysis using a phenomenological approach, as this allowed us to be led by the data when exploring emerging themes related to stakeholder experiences. Our method was characteristic of a small q approach, as we followed the postpositivist framework of qualitative analysis to ensure reliability of the resulting themes related to stakeholder experiences of CVD diagnosis (37). KA used NVivo to conduct the first round of analysis, following the steps recommended by Braun and Clarke (38). This involved first becoming familiar with the data, followed by an initial code generation and theme identification. After this initial round of analysis, secondary coding and review was conducted by OG and AS to validate theme extraction and support in the naming of themes. The findings were discussed between each of the reviewers before a final decision was made regarding themes and sub-themes to be reported.”

d. Small typo: “We used semi-structured focus groups in people living with CVD” should be changed to ‘with people living with CVD”

Thank you for highlighting this, we have made this correction on page 5:

“We used semi-structured focus groups with people living with CVD to generate discussions of shared experiences during their diagnosis journey, and to allow for direct comparisons between a range of diverse medical experiences which may have been missed or different to information that was collected in a one-to-one interview.”

e. If saturation was not assessed, how did you determine your sample size?

We acknowledge that there was not enough information explaining how we determined our sample size, so this has now been expanded on in the study population section on page 6:

“The number of participants recruited for focus groups and interviews was based on the available time for data collection against the wider project deadlines and the research team’s previous experience conducting qualitative research with clinicians (Abdullayev et al., 2023). A maximum of six participants were included in each of the four focus groups to allow adequate time for each participant to share their views and experiences.”

f. Scientific rigour: It’s not clear how presenting the first round of thematic analysis findings to clinicians as a poster increased the transferability of findings to a wider sample. Could you unpack this a bit more? Did you seek feedback from the clinicians? How was this incorporated into subsequent analysis? Similarly, how did feedback from the READ advisory group and the PPI contributor influence the findings?

We apologise for not providing enough detail on our use of PPI and public knowledge exchange to increase the transferability of our findings. We have provided significant expansion on this section on pages 8 and 9, we hope this is satisfactory:

“To increase scientific rigour of our findings, the results of the first round of thematic analysis were presented to clinicians in the form of a research poster at the British Cardiology Society conference to increase transferability of our results to a wider sample. A QR code was provided next to the poster allowing clinicians to scan it and provide their reflections on whether we captured their experiences or comment on what was missing. Those unable to scan the code (e.g. didn’t have a mobile available on hand) provided verbal feedback to the research poster presenter (KA). Feedback from 5 clinicians was integrated into later stages of analysis.

We also consulted with the London-based NIHR Maudsley Biomedical Research Centre’s Race, Ethnicity and Diversity (READ) advisory group to provide further cultural insight on our preliminary findings, which were presented via a series of presentation slides summarising the key findings so far. Verbal discussions were facilitated by the lead researcher (KA) and written up to be discussed within the research team and incorporated within later stages of analysis.

Neither form of cross validation resulted in major changes to the analysis, however it supported the organisation and description of the themes and subthemes reported. While it is not possible to remove the subjective bias of the researchers conducting the analysis, this PPI-led approach to thematic analysis increases the credibility of our findings, which ultimately increases its translatability beyond our sample.”

3. Results

a. I think that Table 1 should sit within the results, not methods. It would be useful to know more about the roles of the clinicians, e.g. were they based in primary or secondary care?

We have relocated Table 1 so that it sits within the results on page 10 and we have included an extra column to include information on the proportion of primary vs secondary vs emergency care clinicians included in our sample.

b. And did you collect data on how long it was since a patient’s CVD diagnosis? This might be useful to consider when interpreting their recollections of being diagnosed.

Unfortunately, we did not collect information on how much time had passed since the patient’s CVD diagnosis, although this was sometimes shared throughout the focus groups. We agree with the reviewer that this would have been useful information to have recorded, and so we have decided to add this is a limitation of the study in the discussion on page 26:

“Finally, we did not collect data on when participants were diagnosed with their cardiovascular disease. This information could have been useful to understand how lived experiences varied for participants who were diagnosed more recently compared to those who were diagnosed decades ago. There also could be greater recollection bias from participants who were describing experiences from a long time ago, which undermines the quality of evidence. Future studies exploring clinical experiences of diagnoses could specify a cut-off date during recruitment to avoid this potential bias in the data.”

c. How many ppts were in each focus group?

We apologise for omitting this information, it has now been added in the procedure on page 7:

“The focus groups and interviews follow a pre-approved, semi-structured question schedule, split into two sections (Appendix 2). Each focus group included either five or six participants. All focus groups and interviews were conducted online using Zoom (<https://zoom.us>), with focus groups lasting about 90 minutes and interviews ranging between 30-90 minutes, based on clinician availability.”

d. I did not feel very confident in the analysis when reading the findings, as quite often the description and interpretation of a quote did not seem to match the quote itself. Perhaps it would be worth revisiting the description of the themes to consider how the quotes are interpreted? I've given some examples from the first theme below:

- The quotes on page 11 “is this AF or am I just getting myself worked up with it? So, you kind of doubt your reliability, don't you?” (P18) doesn't in itself show that it was confusing symptoms that gave the patient doubts. It suggests more that they were unsure whether they were overreacting or not.

- The quote on page 12 para 1 (“their job seems to be more physical in terms of, you know, treating the condition rather than, you know, the mental aspect of it or the ongoing aspects of it” (P29)) seems to be about the patient's perspective of whether the clinician takes a holistic approach to treating their condition, not diagnosing it, which is the focus of the research question.

We agree the exact interpretation of these statements is not possible to be definite about. However, our clinical experience of treating hundreds of patients with intermittent AF leads us to interpret the first statement as the person is not sure whether or not their symptoms truly relate to an underlying rhythm abnormality such as an episode of AF, rather than an over-reaction, either to the symptoms or to the previously diagnosed health condition. While the second example provided speaks to patient concerns with clinicians not taking in factors beyond their physical health symptoms when making decisions regarding their healthcare pathway. As with all qualitative research, the subjectivity of the researcher cannot be removed from the analysis, therefore while we acknowledge the reviewer's perspective, we have chosen to keep the interpretation of the quotes the same in these instances.

- The second para on page 12 also felt a bit confused to me. The patient quote (“whenever I went to my GP and said, I'm not feeling well, he would say, well, you have a heart problem but I'm pretty sure that the last two years I've been suffering from something other than my heart, or in addition to my heart”) suggests they were already diagnosed as having a heart problem but felt something else was wrong, so is this about being diagnosed with CVD? The following clinician quote that “chest cardiac symptoms can sometimes be very vague and overlap with other diagnoses” (CL9) is interpreted by the authors as “contributing to false attribution of non-cardiac symptoms to a cardiovascular disease”, but isn't it the other way around? I may have misunderstood this but suggest it needs unpacking a bit more to make it clearer for the reader.

However, we do agree that these statements also point to the fact that a person's symptoms may relate either to co-existent cardiac and non-cardiac problems and there is potential clinical error in their attribution either to a known heart condition, or to an alternative diagnosis. As the reviewer suggests we have clarified this further in the revised manuscript on page 13:

“This problem was echoed by clinicians, who acknowledged that “chest cardiac symptoms can sometimes be very vague and overlap with other diagnoses” (CL9), contributing to false attribution of non-cardiac symptoms to a cardiovascular disease, or vice versa.”

- Para 3 on page 12 is confusing as it aligns a patient's intuition ("knew something wasn't right") with clinician's holistic approach to applying their knowledge ("knowledge of the biology, physiology, pharmacology, pathology, and histopathology, and as well as clinical knowledge"), which seem very different things.

We appreciate that our interpretative comparison of these two quotes was not coherent and therefore have made a slight adjustment to enhance clarity and better represent our interpretation that both patients and clinicians rely on their personal subjective experience when making decisions regarding their diagnosis or symptom interpretation on page 13:

"Both clinicians and patients described experiences related to intuition and previous experience guiding decisions regarding their symptom interpretation, with one patient sharing how they "just woke up and knew something wasn't right and when the ambulance came and they said, what's wrong? And I said, I don't know but something is" (P20) and a clinician describing how they depend on their "clinical knowledge" (CL1) to help them interpret a patient's history, in addition to "biology, physiology, pharmacology, pathology, and histopathology"."

a. Theme 2 quickly moves from discussing diagnosis in the opening sentence, to discussing involvement in managing the condition which is not focus of the research question.

Thank you for highlighting this inconsistency, we have clarified our interpretation of Theme 2 to show how patient characteristics, especially in relation to self-efficacy and autonomy, influence patient behaviours in relation to their CVD on page 14. We would also like to share the difficulties associated with distinguishing diagnosis and long-term management, as many patients self-manage their symptoms before receiving a diagnosis, while others receive several diagnoses over a period of months or years. We have attempted to address this complexity throughout the study as best as we could within the constraints of the journal's formatting requirements.

"These differences manifested in several ways, for example in the level of autonomy patients adopted when providing information regarding their health, with one individual describing how their GP stopped "following up, asking for my readings, so I just stopped doing it." (P10), while another patient "just took myself to the GP" (P28)."

b. Table 2 and the frequent references in the following para and throughout the discussion about the 'size' of each theme or 'weight of evidence' feels a very quantitative approach, which again is not compatible with reflexive thematic analysis. Number of mentions of a theme doesn't necessarily indicate how important it is. Perhaps a coding manual with definitions of each code and an example quote would be more useful here, like an abbreviated version of the theme table in your appendix?

We appreciate your feedback on the presentation on our findings. Our decision to present the size of themes was influenced by two key drivers: firstly, our PPI advisor (LL) suggested more emphasis to be placed on in-text description, as they felt presenting a codebook within the results section would undermine the depth of each sub-theme, therefore we opted to provide more interpretative description throughout the results section instead of having a table with definitions of each code. Secondly, we wanted to structure the paper in a way that would be most familiar to the readership of the BMJ Open journal which is mainly comprised of clinicians and clinical researchers, as opposed to heavily qualitative practitioners or researchers. We hope that changes made to our data analysis description also reduces the confusion as we have clarified that our approach was more characteristic of a post-positivist framework as opposed to a reflexive thematic analysis. We hope the reviewers find this adequate.

4. Discussion

a. Page 20: The suggestion that “The growing interest in the implementation of personalised healthcare via wearable devices and digital medicine provides opportunity to account for these differences and improve patient experience and health outcomes”, feels a sweeping statement, given that the theme around patient characteristics covers diverse patient experiences including anxiety around diagnosis, lack of social support, and clinician perceptions around patients’ SES, which digital devices cannot be assumed to address. More nuanced consideration of the possible benefits and issues of using digital devices in this context would be good here.

We agree that we were too simplistic and sweeping in this statement, so in the discussion, we have acknowledged the fact that digital technologies represent a balance of more flexibility in the timing, and pace of obtaining information but this may be at the expense of a more nuanced and individualised consultation. Please see the changes made on page 24:

“Although current healthcare has significant limitations, it does at least have the theoretical capacity for the clinician to tailor some (but not all) aspects of the medical encounter to the needs, understanding, and preferences of the patient. However, it is clear that digital tools without the capability to be customised to the individual user (either patient or clinician) run the risk of failing to adapt to these patient differences. It is important for such digital technologies to be designed to be adaptable enough to account for such differences. Conversely, the ability to provide data where and when the patient feels most comfortable at their own pace, and to do so outside of a potentially stressful medical encounter, may provide opportunities to account for these differences. Ultimately this could improve patient experience and health outcomes in ways that would not be possible in the traditional healthcare pathway which has fixed times and locations in which data is obtained (44,45).”

b. The considerations in Table 3 feel very broad, e.g. “Make future solutions more inclusive for patients of different ages, literacy levels, mental and physical health conditions”, is not particularly focused on digital interventions in this context nor taking account of the implementation barriers to achieving these.

Thank you for this feedback. We agree that the table of considerations in our original manuscript did not focus on the applications to digital interventions in the context of our findings, so we have now added an additional column labelled ‘Potential Design Aspects of Digital Twin Technologies’ that expands on how the considerations can be applied in practice. Please see page 23 for the new version of this table.

c. Might the study have been strengthened by including some patients with ambiguous symptoms who have not yet been diagnosed, or only including patients who are currently undergoing diagnosis?

Thank you for this suggestion. We have added this insightful point to our discussion on page 26:

“Moreover, further investigation could be done to determine whether the present study’s findings are consistent with patients currently undergoing diagnosis, especially since the COVID-19 pandemic, as this may highlight the most urgent areas that would benefit from novel digital health technologies.”

d. Further consideration about the impact of the recruitment approach would be useful to see, e.g. why was twitter used rather than a less academic social network like facebook? Was there any option to sign-up without an email address? From Figure 1 it looks like 1 person was recruited via social media rather than via the existing research groups of CVD patients agreeing to be invited to further research –which might be useful to reflect on? It would also be useful to include the email that was used to contact people initially to ask if they were interested, after they got in touch? Could you also

report if people were paid to take part? What steps could have been taken to make it easier for people to take part, e.g. option to do an interview via phone call instead of the Zoom focus group for non-Zoom users, or those who would prefer a 1:1 discussion? You lost 7 participants who were unavailable at the focus group times, but they could have been included had 1:1 interviews been offered. What happened to the 10 people who you lost contact with after they had agreed to take part?

Thank you for providing such a thorough recommendation regarding our recruitment methods. We would like to clarify that both Twitter and Facebook were used for recruitment (page 6,7), as Twitter has a wider potential reach as it can be used to make public posts, while Facebook was used to post recruitment posters within known community groups, thus both provided different advantages. Unfortunately, it was not possible to sign up to our study without an email address as this was vital for communication during the set-up of the focus group and following participation for payment. We acknowledge this as a limitation when conducting online qualitative studies and have included it as a limitation of the study at the beginning of the manuscript and in the discussion, highlighting our awareness of how our recruitment methods may have produced bias on page 26:

“The use of online recruitment platforms and snowball convenience sampling to recruit our participants may have produced a biased sample of individuals who are more involved in their own healthcare and new technological developments in cardiovascular area. Therefore, our sample may be less representative of patient and clinician populations who are less digitally literate who may face even greater challenges in receiving or delivering accurate and efficient CVD diagnosis. Future research should consider ways to include more seldom heard groups in research investigating contributors to delayed and inaccurate diagnosis of CVD.”

Unfortunately, it was not possible to gather information on why contact was lost with certain participants as they did not respond to three rounds of follow up emails. Finally, our description on page 5 illustrates the benefits of conducting a focus group compared to an interview that informed our decision to not provide an interview option for lived experience participants:

“We used semi-structured focus groups with people living with CVD to generate discussions of shared experiences during their diagnosis journey, and to allow for direct comparisons between a range of diverse medical experiences which may have been missed or different to information that was collected in a one-to-one interview.”

The following addition has been made based on the reviewer’s recommendations on page 7:

“Participants were compensated for their time with a £25 Amazon voucher.”

e. For the clinician interviews, where did the personal connections come from? It was interesting that 6 of the 9 clinicians had completed more than 20 years in the service. How might it have influenced the findings that very few were new to the role?

Thank you for bringing this potential bias to our attention, we have added a further comment on this in our discussion on page 22:

“Although, the majority of the clinician sample was highly experienced therefore this may have introduced a bias in their perspectives related to barriers, for example their years of experience may have made them more or less affected by certain issues. Future studies should ensure a variety of levels of clinical experiences are considered to prevent potentially biased interpretations. Nonetheless, the contribution of PPI groups to the design, recruitment, and analysis process, and

cross validation of preliminary findings with a range of clinicians at the British Cardiovascular Society Conference increase the transferability of our findings.”

5. Conclusion – The conclusion feels more of a quick wrap-up statement. Perhaps more consideration could be given to what are the really interesting and unique findings from this research in relation to using digital interventions for CVD diagnosis?

We acknowledge that our conclusions in the original manuscript did not do our unique and interesting findings justice, therefore we have added a few additional sentences to highlight how our results can be translated into useful recommendations for novel technologies on page 27:

“Several considerations have been suggested that would inform development of digital twin and other technological innovations to improve the accuracy and efficiency of CVD diagnosis. Such technologies must overcome key barriers related to time, patient-clinician communication and difficulties tailoring to individual patient differences within the diagnosis pathway. Successful innovations need to increase efficiency, improve patient-clinician communication, and to provide a tailored approach to diagnosing individuals with heart disease. While this study provides insight into patient and clinician experiences of these challenges, further research is required to enhance our understanding of how these experiences differ between ethnic groups and genders.”

Reviewer: 2

Dr. Emma Kirby, University of New South Wales

Comments to the Author:

Thank you for the opportunity to review this study, which aims to better understand delayed and inaccurate diagnosis of cardiovascular disease from perspectives of both patients and clinicians. The study and findings have much to offer, not only in relation to CVD, but in terms of the various subjective positions and perspectives that coalesce to shape the experience of delay in diagnosis. As such, the findings could be most useful in thinking more broadly about how patients/people interpret symptoms, patient-clinician relationships, and how the everyday bureaucracies and challenges within health systems can shape outcomes (indeed, some of these ‘bigger’ implications could, and in my view, should, be engaged with in the Discussion section, as they would further extend the relevance and potential impact of the research).

Thank you for this insight. We have added a section at the end of our discussion under the subheading of ‘Implications for Digital Technology Development’ on page 26-27. This section brings together the bigger implications of our findings in relation to novel health technologies. We hope you find this sufficient.

“Several considerations have been suggested that would inform development of digital twin and other technological innovations to improve the accuracy and efficiency of CVD diagnosis. Such technologies must overcome key barriers related to time, patient-clinician communication and difficulties tailoring to individual patient differences within the diagnosis pathway. Successful innovations need to increase efficiency, improve patient-clinician communication, and to provide a tailored approach to diagnosing individuals with heart disease. While this study provides insight into patient and clinician experiences of these challenges, further research is required to enhance our understanding of how these experiences differ between ethnic groups and genders.

Although our work focussed on diagnosis as the first essential step in a clinical pathway, diagnosis alone does not improve patient outcomes. This requires interventions (such as behaviour change, drug treatment, or surgery) and most CVDs also require some form of ongoing monitoring to detect changes over time and attempt to predict deterioration early enough to prevent complications such as

hospital admission or death. It is likely that many of the characteristics required for better diagnosis could be transferred to technologies used to guide decisions about interventions and monitoring, but this requires further research.”

In my view the article could be suitable for publication, once some important methodological and related issues are addressed. I outline these below – I hope that the authors find these suggestions/comments constructive in advancing the article. Essentially, my comments relate to how the study is situated, and how revisions can improve the alignment of aims with approach, analysis, and interpretation of findings.

1. Some explanation of rationale for focus groups and interviews (pragmatics of data collection are mentioned briefly, but some engagement with the appropriateness of these methods for the type of knowledge required to address research aims/questions is also needed).

Thank you highlighting this omission, we have reflected on the suitability of using qualitative methods to address our research question in the discussion, with an emphasis on the advantage it gave for gaining a deeper insight into stakeholder experiences and perspectives on page 25:

A key strength of the present study is the use of qualitative interviews and focus groups to achieve its objectives, as this allowed for open ended questions to provide a more in-depth understanding of the current challenges faced by stakeholders in diagnosis for CVD. Specifically, the decision to carry out focus groups with lived experience groups allowed for lively discussions where participants felt more comfortable to disclose their personal experiences and relate to their peers.”

2. Relatedly – the objectives of the study, as they stand, are not in and of themselves obviously aligned to a qualitative approach – so some justification of approach (relative for example to a survey of patients and clinicians) is needed. As a note – the objectives stated in the previously published protocol have better alignment with a qualitative approach; however, there is also not explicit justification or rationale for this approach (methodologically, epistemologically, theoretically) in the protocol – so the protocol cannot be relied on in isolation to provide the required information.

We completely agree with reviewer 2 on all accounts and have made an effort to better align our report with the qualitative methodology that we have conducted for the study. Firstly, we have expanded on our description at the end of the introduction on page 4 to better introduce our adapted objectives that have been written in a similar tone to the previously mentioned protocol. We also added greater emphasis in the study design section on the intentional decision to carry out a qualitative study to address the research question on page 5:

“Digital technologies could play a greater role in facilitating CVD diagnosis, but this requires a deeper understanding of stakeholder experiences of CVD diagnosis, to support development of technological solutions which meet these needs. Given the lack of existing evidence about the lived experience of heart disease diagnosis, we took a qualitative approach to investigate the following objectives:

1. Understand the range of challenges faced by stakeholders in diagnosis for CVD.
2. Understand potential discrepancies between patient and clinician experiences of CVD diagnosis.
3. Make recommendations for requirements of future health technology solutions for improving CVD diagnosis.”

“A qualitative approach was taken to capture the depth of experiences and the complex nature of living with a CVD, as this would not be easily achieved using quantitative approaches.”

3. The main issue in need of addressing for the article to be suitable for publication as a qualitative study is (the current lack of) engagement with a theoretical or conceptual framework. Reflexive thematic analysis as an approach to qualitative analysis requires explicit consideration of theory, and/or explanation of the philosophical, ontological, and epistemological underpinnings of the approach to data and knowledge production. An approach to analysis is not sufficient; an approach to research methodology and an approach to understanding the knowledge that will then be generated is also required. At present, the article is an (albeit very well done) example of more proceduralist approaches to qualitative work – given that Braun and Clarke’s reflexive thematic analysis was drawn on within the study, some inclusion/integration of how a theoretical or conceptual framework guided the research is needed. Braun & Clarke in their more recent work have published on this – engaging with the issues outlined in their below articles would be useful (particularly in terms of rigour of process and approach, as distinct from the issues of scientific rigour currently included in the article).

<https://www.tandfonline.com/doi/full/10.1080/17437199.2022.2161594>

<https://www.tandfonline.com/doi/full/10.1080/26895269.2022.2129597>

As previously mentioned in our response to reviewer 1, we are incredibly grateful for the efforts both reviewers have gone through to support us in maintaining high quality qualitative research. We hope that our clarifications on page 8 are deemed sufficient:

“We conducted an inductive thematic analysis using a phenomenological approach, as this allowed us to be led by the data when exploring emerging themes related to stakeholder experiences. Our method was characteristic of a small q approach, as we followed the postpositivist framework of qualitative analysis to ensure reliability of the resulting themes related to stakeholder experiences of CVD diagnosis (37). KA used NVivo to conduct the first round of analysis, following the steps recommended by Braun and Clarke (38). This involved first becoming familiar with the data, followed by an initial code generation and theme identification. After this initial round of analysis, secondary coding and review was conducted by OG and AS to validate theme extraction and support in the naming of themes. The findings were discussed between each of the reviewers before a final decision was made regarding themes and sub-themes to be reported.”

4. Relatedly, the extent of interpretation of the data could be improved. Part of the problem here does not lie with the authors – I want to be clear in acknowledging that journal formatting guidelines, including in BMJ Open, often serve to stifle attempts to show interpretation (e.g., through separation of Results and Discussion sections). That said, some more emphasis on the subjectivities of interpretation of findings in relation to existing theory/concepts – and not just in relation to empirical findings from other studies – would improve the contribution of the article. This will also improve the alignment of the results and discussion with the approach/practices of reflexivity that is an important part of reflexive thematic analysis.

We were unsure whether the reviewer was referring to the theories and concepts of the thematic analysis or of heart disease diagnosis. As there are no background theories or concepts in relation to the latter, we adjusted our scientific rigour section to acknowledge the role of researcher subjectivity and how this was dealt with in our analysis on page 9:

“Neither form of cross validation resulted in major changes to the analysis, however it supported the organisation and description of the themes and subthemes reported. While it is not possible to remove the subjective bias of the researchers conducting the analysis, this PPI-led approach to thematic analysis increases the credibility of our findings, which ultimately increases its translatability

beyond our sample.”

5. Other minor issues:

a. Discussion: how does the size of themes align with an RTA approach? Some consideration of this, and amendment accordingly is needed.

We hope that the response to reviewer 1’s similar concerns are sufficient to satisfy the reviewer, as we have clarified that our approach was more characteristic of the coding reliability thematic analysis than reflexive TA.

b. The emphasis within the study on the creation/development of a digital health tool could be further contextualised, particularly in relation to the thematic findings. Alternatively, the authors might explain in the early parts of the article, why the digital tool and lived experiences of engaging with digital health tools are not the focus of the article.

We appreciate this wasn’t entirely clear in the original manuscript, so we have specified that this paper deals with one half of the topic guide questions in the methods section on page 5:

“The question topic guide involved two main parts – clinical experiences and technology-related experiences – however, the present study includes data related to stakeholder perspectives on contributors to delayed and inaccurate diagnosis of cardiovascular disease.”

Yours faithfully,

Kamilla Abdullayev (corresponding author)

Kga21@sussex.ac.uk